

# Development of a full-waveform voltage and current recording device for multichannel transient electromagnetic transmitters

Xinyue Zhang, Qisheng Zhang, Meng Wang, Qiang Kong, Shengquan Zhang, Ruihao He, Shenghui Liu, Shuhan Li, Zhenzhong Yuan

School of Geophysics and Information Technology, China University of Geosciences (Beijing), Beijing, 100083, China
*Correspondence to*: Qisheng Zhang (zqs@cugb.edu.cn)

**Abstract.** Owing to the pressing demand for metallic ore exploration technology in China, several new technologies are being employed in the relevant exploration instruments. In addition to possessing the high resolution of the traditional transient

electromagnetic method (TEM), high-efficiency measurements, and a short measurement time, the multichannel TEM (MTEM) technology can also sensitively determine the characteristics of a low-resistivity geologic body, without being affected by the terrain. Besides, the MTEM technology also solves the critical, existing interference problem in electrical exploration technology. This study develops a full-waveform voltage and current recording device for MTEM transmitters. After continuous acquisition and storage of the pseudo-random, large current signals emitted by the MTEM transmitter, these signals

are then convoluted with the signals collected by the receiver to obtain the earth's impulse response. In this paper, the overall design of the full-waveform recording apparatus, including the hardware and upper-computer software designs, the software interface display, and the results of field test, are discussed in detail.

## 1 Introduction

The full-waveform voltage and current recording device of a multichannel transient electromagnetic (MTEM) transmitter is

an important component of MTEM exploration equipment. This device is responsible for the continuous acquisition and storage of the large pseudo-random current signals emitted by the MTEM transmitter. These signals are then convoluted with the signals collected by the receiver to obtain the impulse response of the ground under test (Zhong and Xue, 2014; Xue et al., 2015). Electrical methods (including conductive and inductive electrical methods) are crucial in metallic ore exploration. The conductivity, dielectric constant, and magnetic permeability of the rock ore are the main physical parameters measured in

electrical geological exploration (Xue et al., 2007; Newman, 1989). Currently, the most critical problem with these methods is the interference from the area being explored. Solutions to reduce the effects of interference include increasing the transmission power of the field source, expanding the distance between the transmitter and receiver (conductive electrical methods), and reducing the observation signal frequency (inductive electrical methods). However, these measures often result in insignificant improvements in the signal-to-noise ratio (SNR) and higher costs. Therefore, they cannot satisfy the demands

of the metallic ore exploration industry (Wright et al., 2001). The device examined in this study uses pseudo-random code to



transmit certain artificial signals; the induced electromagnetic signals are then collected using a receiver at the remote end. Coherent decoding can significantly enhance the SNR of the acquired data and improve the exploration outcome; hence, it is suitable for monitoring high-power signals. Furthermore, the device also uses related technology to ensure the accuracy of the acquired signal amplitude and acquisition time, thereby achieving simultaneous acquisition of the high- and low-speed voltage

and current.

In the future, the full-waveform voltage and current recording device can be used for marine exploration, marine electromagnetic exploration as a supplement to seismic exploration technology, can directly detect oil and gas in the structure of oil and gas, its status in the field of marine exploration is increasing. MTEM is accompanied by marine electromagnetic exploration came into being, if this technology is applied to land and sea oil and gas exploration, will greatly reduce the risk

of three-dimensional seismic exploration, non-seismic methods of oil and gas resources exploration will have a very positive impact.

This study discusses the development of the full-waveform voltage and current recording device for MTEM transmitters. The article is structured as follows: Section 2 presents the hardware circuit design of the full-waveform recording device; Section 3 describes the upper-computer software design and Section 4 presents the test results.

**2 Hardware circuit design of the full-waveform recording devices**

The hardware circuit design of the full-waveform recording device for MTEM transmitters can be divided into the voltage and current channel designs (Ziolkowski et al., 2007). The design principles are depicted in Figs. 1 and 2. The voltage channel divides the input signal into two levels: large-range and small-range. For large-range levels, the maximum input for the transmitter voltage amplitude is 1000 V, whereas, when adjusted to the small-range level, the maximum input is 100 V; relays

are used to switch between the large and small-range levels. As the maximum voltage amplitude emitted by the transmitter is 1000 V, a voltage-reducing follower is needed before being input to the analog-to-digital converter (ADC) for conversion (Boris and Bernhard, 2003). The input-end of the low-speed ADC is a differential converter; hence, the single-ended signal outputs from the follower should be converted into differential signals. However, as the input-end of the high-speed ADC is single-ended, only an adder circuit is needed. In addition, a hysteresis comparator is constructed to achieve frequency

measurement functions (Jia et al., 2009).

The current channel differs from the voltage channel in that it does not require repeated frequency measurements; hence, a hysteresis comparator is not required. The large current signals from the transmitter need to undergo a voltage-current conversion, which is achieved using a shunt. Hence, an amplifier circuit with a high bandwidth and high amplification magnitude is required. The other parts of the conditioning circuit are similar to those of the voltage acquisition.

The signals emitted by the transmitter are strong with large voltages and currents, whereas those of the master FPGA are weak (Huang et al., 2015). Hence, an isolation barrier is needed between the two for protecting the master circuit and the upper-computer connected to it. Therefore, we designed a high-speed optocoupler isolation circuit. Every digital input/output of the



ADC, on the acquisition board, requires a high-speed optocoupler for isolation; the input and output directions of the optocoupler were verified one-by-one to ensure circuit correctness (Jhin et al., 2014).

The acquisition circuit of the full-waveform recording device can be divided into low-speed and high-speed. The low-speed acquisition circuit uses the ADC, ADS1271, to achieve a sampling rate of 32 kSPS. A high-precision 24-bit Δ-Σ type ADC was used, supporting the following modes: high-speed, high-precision, and low-power (Li et al., 2013). Under the high-speed mode, the sampling rate can reach a maximum of 105 kSPS. In the high-precision mode, the SNR can reach 109 dBm and in the low-power mode, the overall power consumption is 35 mW (Texas Instruments,2007). For high-speed acquisition, ADC,

AD9226, was used to achieve a sampling rate of 40 MSPS; this is a 12-bit pipeline ADC with a maximum sampling rate of 65 MSPS and a bandwidth of up to 750 MHz. For an input signal of 31 MHz, the spurious-free dynamic range is 85 dM (Analog Devices, 2001).

The FPGA master circuit is the control core of the MTEM transmitter full-waveform voltage and current recording (Sun et al., 2016). It is responsible for the core functions of the recording device, including control acquisition and data transfer

(Ziolkowski et al., 2010). The master circuit is mainly composed of the FPGA and a few external storage devices; its block diagram is depicted in Fig. 3. The storage devices include an SRAM, SDRAM, and serial and parallel flashes. Among them, the SDRAM is used as the data buffer for low-speed acquisition and the parallel flash, as the memory for the FPGA-configuration firmware. In order to ensure data integrity in the low-speed acquisition, an external SDRAM was used as a buffer for the data from this circuit. The 256-Mbit single-clock SDRAM used in this study has four groups of internal banks. Each

bank has four trillion stored words and each word has a 16-bit length; i. e., the data bus is composed of D0–D15 lines. The 80-MHz clock for the SDRAM is provided by the FPGA to satisfy the low-speed acquisition data rate. The master circuit uses high-speed USB ICs to implement the USB transfer protocol, which is simple and convenient. The FPGA of the master circuit uses a JTAG chain to burn the SRAM configuration file onto the serial flash, EPCS16. After the FPGA is powered up again, in the active serial (AS)-mode, it reads the configuration file in the external serial flash internally and completes the IC

configuration process.

After the upper-computer receives the transmission signal waveform, it calculates the voltage and current peak values, frequency, timestamp, and the other transmission waveform information (Zhong et al., 2016). This information is then transferred to the lower computer via the command control channel. The lower-computer forwards this data, as per the serial communication protocol, to the 485 and Bluetooth modules; the data is then transferred to a remote PC via wired or wireless

methods. Fig. 4 shows the process block diagram.

The MTEM method requires the simultaneous recording of the transmitter current and receiver voltage signals followed by the convolution of the two, for calculating the earth's impulse response. Hence, the acquisition time for these two signals is an important parameter (Wright, 2003). This design uses an atomic clock as the clock source for the master circuit and all the ADC clocks are based on stable clock-frequency-dividers or multipliers of the atomic clock outputs (Xu et al., 2004; Zhang et

al., 2015). By synchronizing the pulse per second (PPS) of the atomic clock and the GPS, and then synchronizing the PPS of





the receiver and GPS, we could approximate the simultaneous acquisition of the two acquisition systems (Olalekan and Di, 2015).

## 3 Upper-computer software design

The upper-computer program was built on the Visual Studio platform using C# Windows forms. The program mainly includes the Main Window form (Form1), the Sub Window form (Form2) and a self-defined class function. The program framework is described below: In the main Window form, the read and write functions of each channel were assigned to eight corresponding threads. Sampling or interception was performed on the data flags and channel types to obtain data points for plotting the waveform; four drawing threads were opened to plot the waveforms. Class functions were formed by packaging each channel of the USB equipment into a class, based on the endpoint address. Each class included the endpoint parameters and two methods to read (using the asynchronous read mode) and write data into the files. The read and write methods were assigned to the threads of their own channels, in the main function. The Sub Window form was used to open and review the stored file data. The entire software uses the Metro style, which provides a more humanistic human-computer interface. Fig. 5 displays the Main Window form.

## 4 Test results of the full-waveform recording device

Overall performance testing was conducted after developing the full-waveform recording device. For the low-speed voltage, 20-Vp, 100-Hz sinusoidal signals produced by a signal generator were fed directly to the 1000-V high-voltage input end. Using a voltage-regulated power supply and at a sampling rate of 32 kSPS, 32000 samples were acquired; further, fast Fourier transform (FFT) was performed on these samples and MATLAB was used to generate the time and frequency domain waveforms. For the high-speed voltage, the acquisition performance at the 100-V level was tested. 20-Vp, 100-kHz sinusoidal signals from a signal generator were fed directly to the high-speed voltage input end and the device was powered by a lithium battery (Wang et al., 2015). 4096 samples were acquired at a sampling rate of 40 MSPS; FFT was then performed on these samples and MATLAB was also used to generate the time and frequency domain waveforms. When testing the low-speed current acquisition board, the level was shifted to 50 A; 150-mVpp, 62.5-Hz sinusoidal signals from a signal generator were fed directly to the low-speed current input end; further, MATLAB was used to generate the time and frequency domain waveforms. For testing the high-speed current acquisition board, the level was shifted to 50 A; 150-mVpp, 100-KHz sinusoidal signals from a signal generator were fed simultaneously to the high-speed and low-speed current input ends; then, MATLAB was used to generate the time and frequency domain waveforms (W. W. Zhang et al., 2016).

### 4.1 Low-speed voltage channel

Figure 6 shows the test results of the low-speed voltage acquisition channel. Figure 6a is the time domain waveform of the sinusoid acquired by the recording device, whereas, Fig. 6b is the frequency domain waveform. As seen from Fig. 6b, the

corresponding amplitude of the input 20-Vpp sinusoidal signal was -42 dBFS, i. e., the full amplitude was approximately 2000 Vpp. The SNR at this point was also higher than 50 dB; hence, the overall SNR was higher than 90 dB. A low-speed acquisition precision of up to ±1 % was attained.


### 4.2 High-speed voltage channel

Figure 7 displays the test results of the high-speed voltage acquisition channel. In Fig. 7b, harmonic distortion at an amplitude of approximately -71 dB is observed; hence, its amplitude input was approximately 200 Vpp (200 Vpp is equivalent to -20 dB). Therefore, the overall SNR was higher than 60 dB. A high-speed acquisition precision of up to ±1 % was attained.


### 4.3 Low-speed current channel

Figure 8 shows the test results of the low-speed current acquisition channel. We can see from the amplitude-frequency characteristics that there was a 50-Hz power frequency interference at -73 dB; relatively, many harmonic components were present, the average noise threshold was lower than -100 dB and its SNR higher than 60 dB.


### 4.4 High-speed current channel

Figure 9 displays the test results of the high-speed current acquisition channel.

### 5. Conclusion

In this study, the development of a full-waveform voltage and current recording device for MTEM transmitters was presented.
First, the hardware circuit was designed to complete the pre-conditioning and acquisition of the signals, which were then transferred to the FPGA for data processing. Then, the upper-computer software was designed to further process the data and present it as graphic plots. The overall performance was tested, with individual tests for the high- and low-speed channels for the voltage and current. The maximum transmission current and voltage acquired by the device were 50 A and 1000 V, respectively; a ±1 % sampling precision was attained for both current and voltage. This full-waveform recording device can be
used to monitor the high-power, full-waveform voltages and currents of MTEM transmitters. This recording device has higher precision, finer edge details, lower noise, and other advantages. Hence, it can be used for real-time recording and transmission to the receiver for coherent demodulation.

### 6. Acknowledgements

This work was supported by the Fundamental Research Funds for the Central Universities of China (No. 2652015213), the National Natural Science Foundation of China (No. 41574131), and the National Major Scientific Research Equipment Research Projects of China (No. ZDYZ2012-1-05-01).



**Competing interests**

The authors declare that they have no conflict of interest.

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

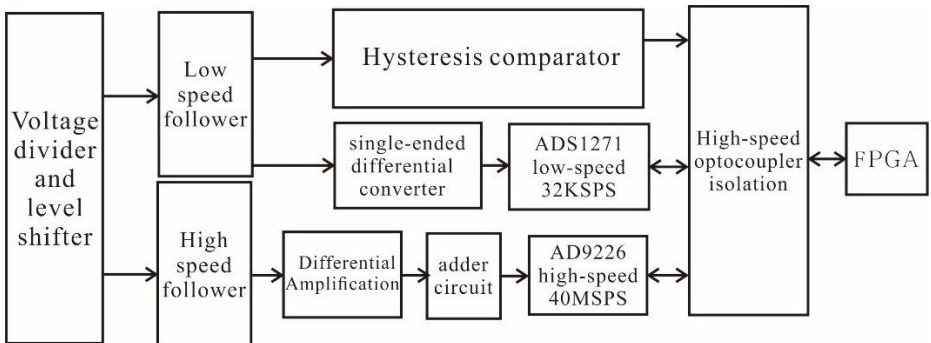

**Figure 1: Block diagram of the conditioning process for voltage acquisition.**

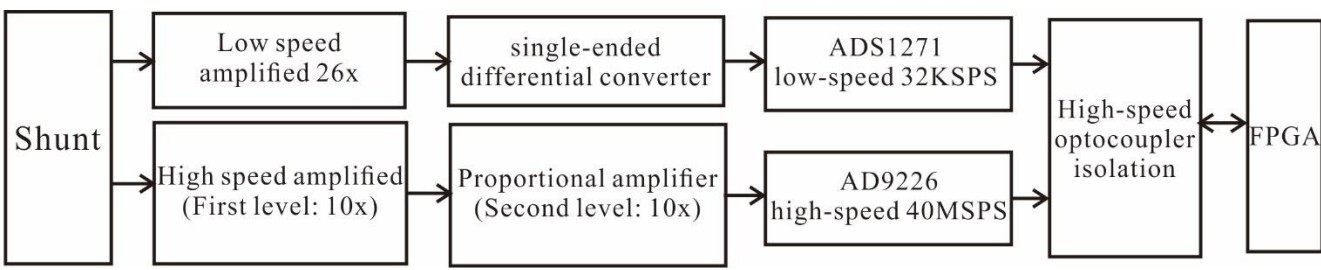

**Figure 2: Block diagram of the conditioning process for current acquisition.**





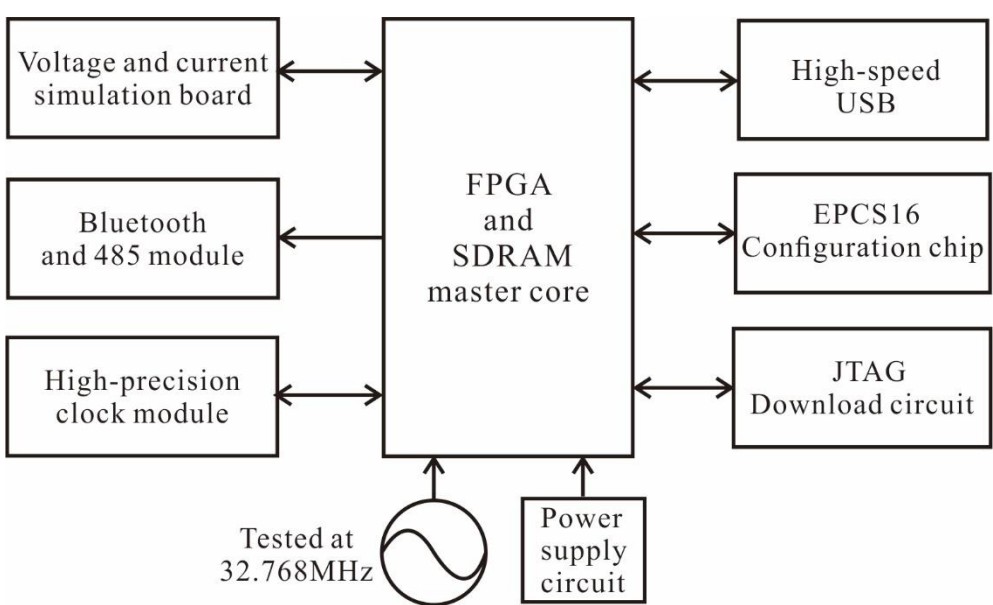

**Figure 3: Block diagram of the FPGA master circuit.**

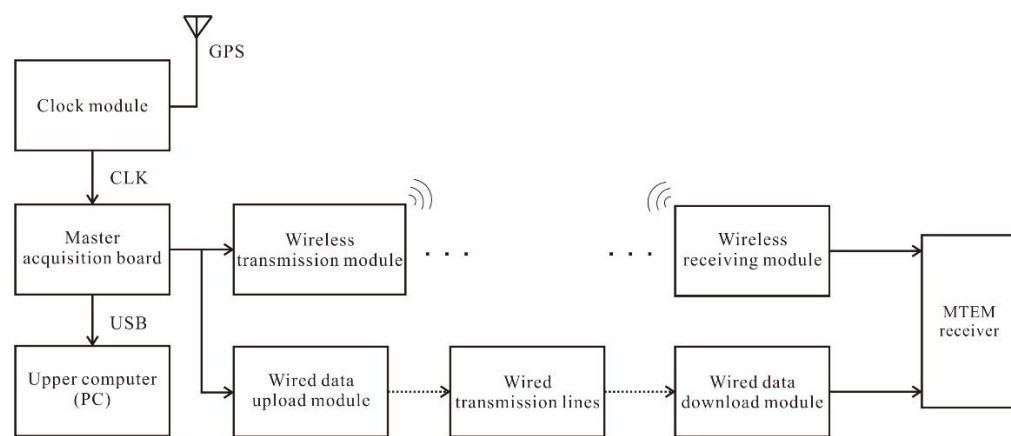


**Figure 4: Block diagram of the Bluetooth and 485 module transmission.**



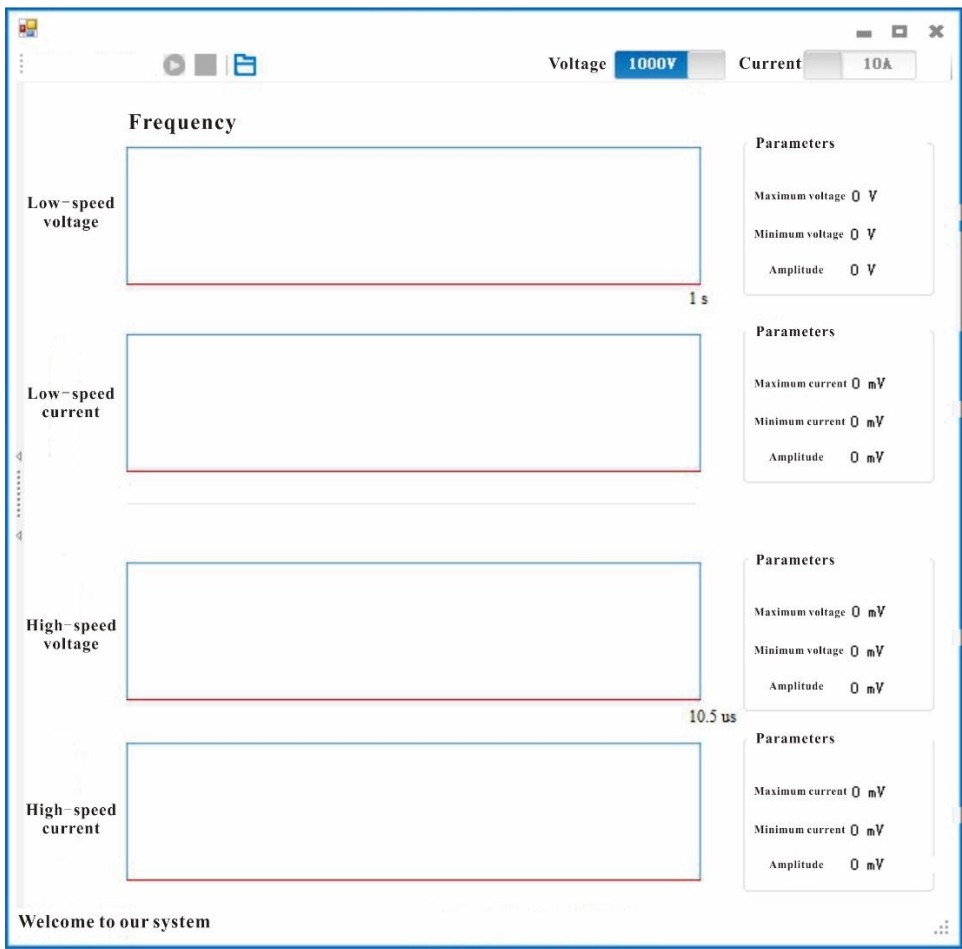

**Figure 5: Interface display of the Main Window form on the upper computer.**




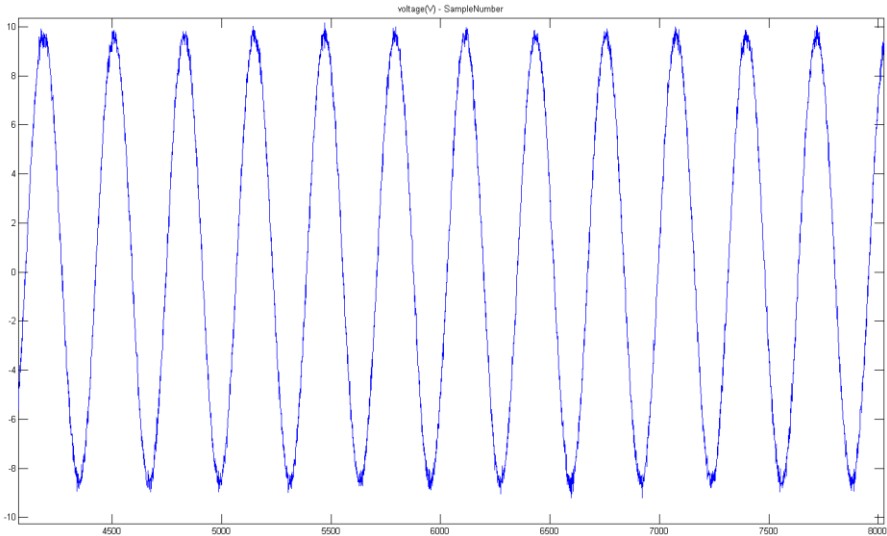

**a. Time domain waveform of the 20-Vpp, 100-Hz sinusoid acquired by the recording device at a 1000-V level**

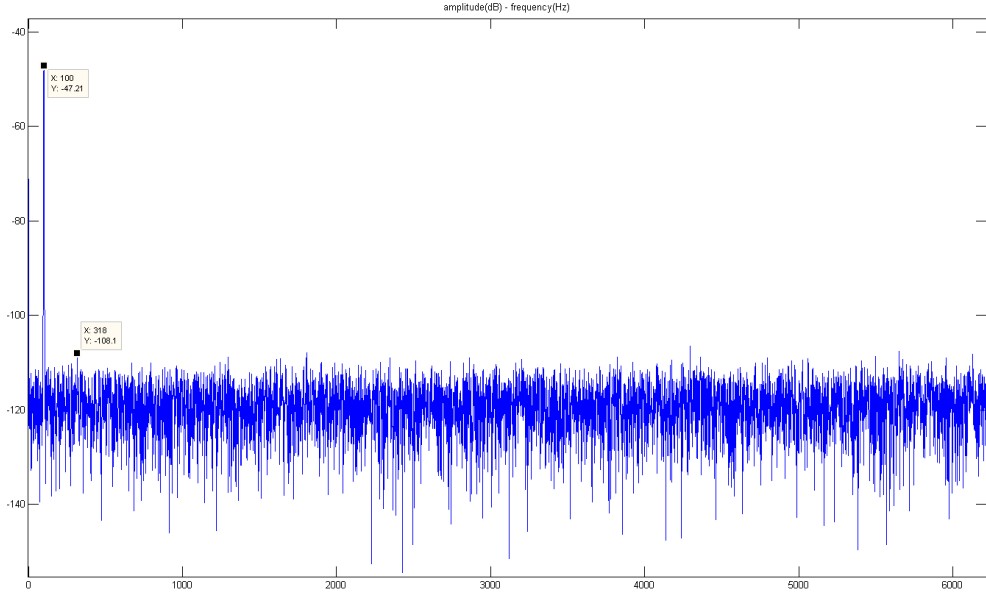

**b. Amplitude-frequency characteristics of the 32000 samples subjected to FFT (Hanning window truncation)**

**Figure 6: Test results of the low-speed voltage acquisition channel.**



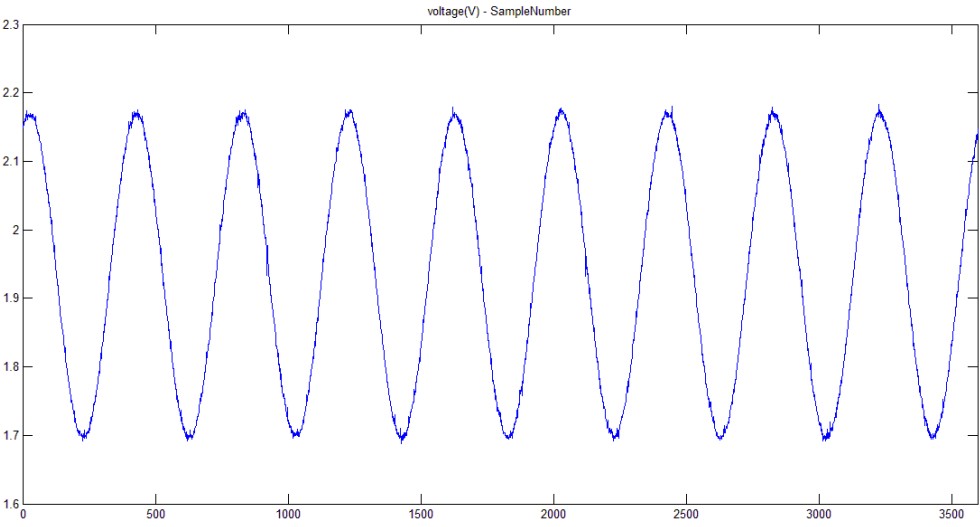

**a. Time domain waveform of the 20-Vpp, 100-kHz sinusoid acquired by the high-speed voltage recording device at a 100-V level**

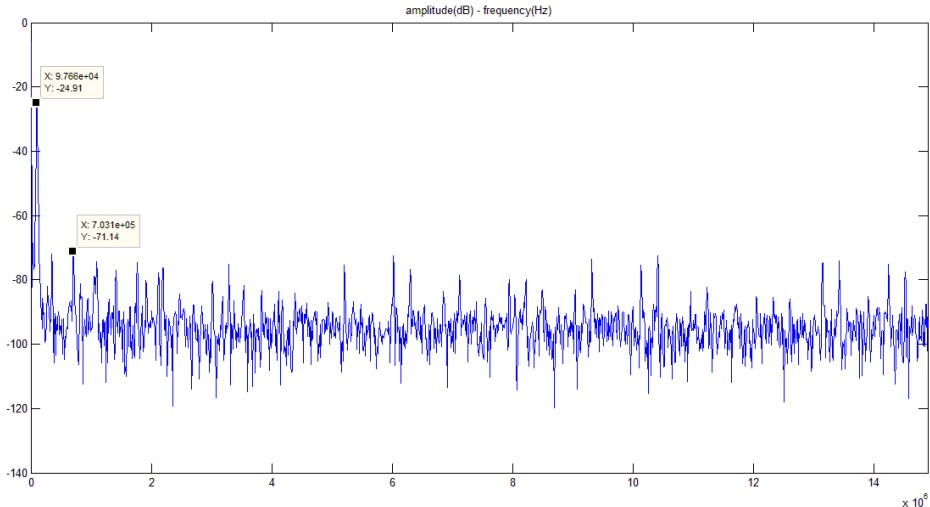

**b. Amplitude-frequency characteristics of the 4906 samples subjected to FFT (Hanning window truncation)**

**Figure 7: Test results of the high-speed voltage acquisition channel.**


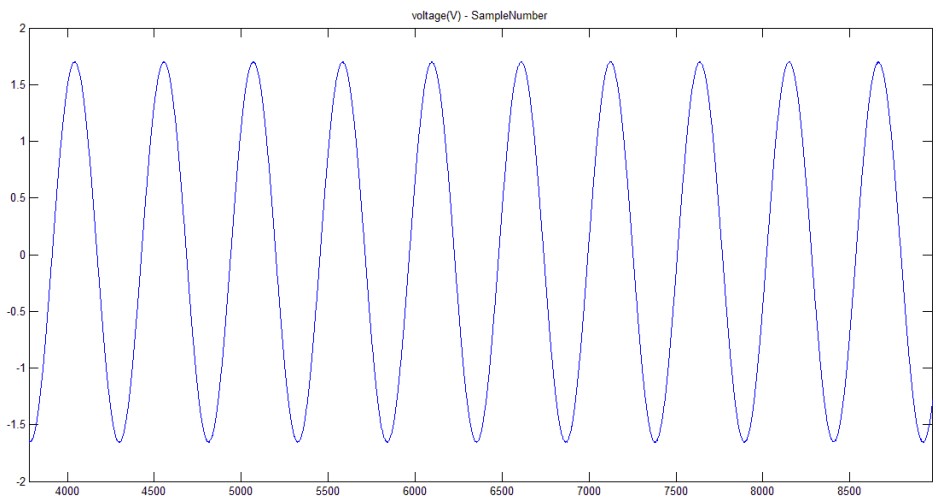

**a. Time domain waveform of the 150-mVpp, 62.5-Hz sinusoid acquired by the low-speed current recording device at a 50-A level**

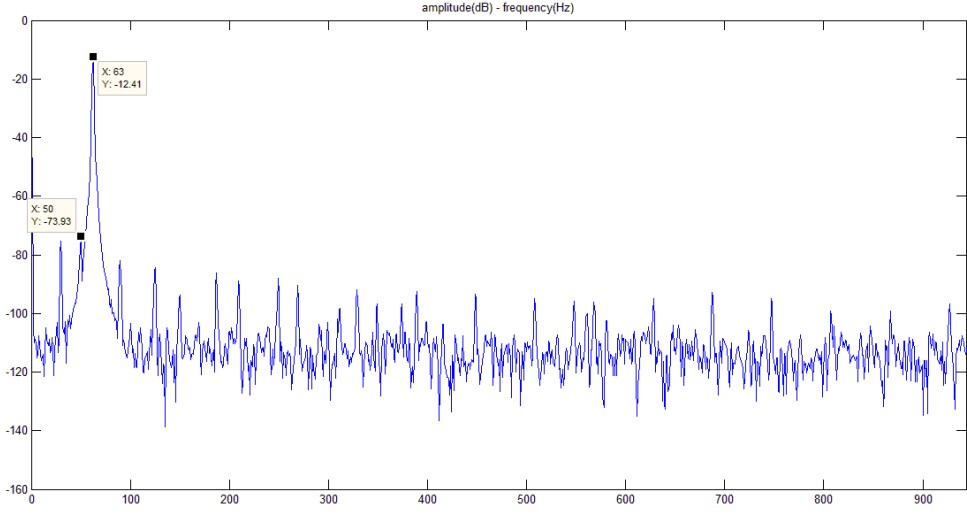

**b. Amplitude-frequency characteristics of the 32000 samples subjected to FFT (Hanning window truncation)**

**Figure 8: Test results of the low-speed current acquisition channel.**





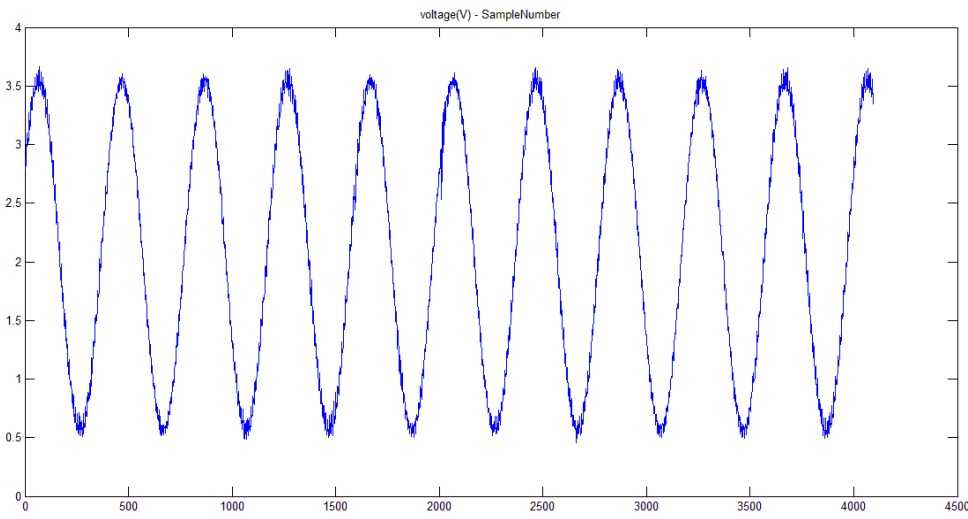

**a. Time domain waveform of the 150-mVpp, 100-KHz sinusoid acquired by the high-speed current recording device at a 50-A level**

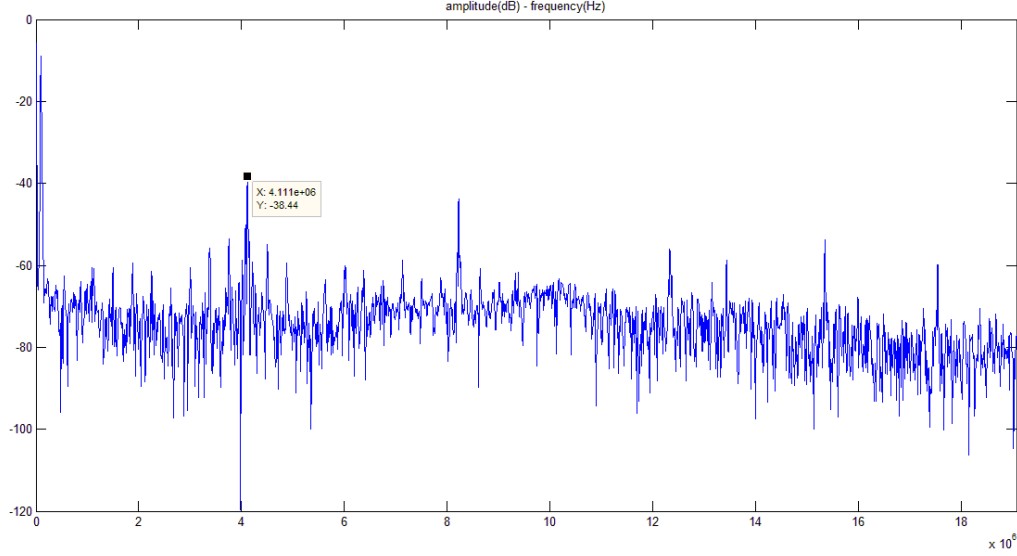

**b. Amplitude-frequency characteristics of the 4906 samples subjected to FFT (Hanning window truncation)**

**Figure 9: Test results of the high-speed current acquisition channel.**