# Peer review of "Development of a full-waveform voltage and current recording device for multichannel transient electromagnetic transmitters"

_Geoscientific Instrumentation, Methods and Data Systems, 2017_

## Referee Comment (RC1) · Anonymous Referee #1 · 11 Jun 2017

First of all, this paper addresses relevant scientific questions within the scope of GI and presents novel concepts and ideas about the development of a full-waveform voltage and current recording device for multichannel transient electromagnetic transmitters. The title clearly reflects the contents of the paper and the abstract provides a concise and complete summary. This paper outlines the scientific methods and assumptions clearly, then it reaches substantial conclusions, the results in discussion part are sufficient to support the interpretations and conclusions. The test results are sufficiently complete and precise. The number and quality of references are appropriate. The overall presentation is well structured and clear. But I still have some questions, so minor revisions should be made. 1. Line54: Please elaborate on what is hysteresis

comparator. 2. Line61: Please explain how you implement the optocoupler isolation. 3. Line91: According to your discussion, the atomic clock is a very important part in your device, so in order to help the author better understand the principles of your instrument, it is necessary to elaborate on its circuit structure.

After reading through the full paper, I find the language is fluent and precise.

---

## Short Comment (SC1) · 20 Jul 2017

To esteemed anonymous referee #1. Your comments impress us a lot. We really appreciate your time and energy for reviewing our manuscript. It is hard for us to express our grateful feeling. In fact, we have learned many things during this revision process, and such experience would be very helpful for our future study. We herewith provide our response to your comments as below: 1. Line54: Please elaborate on what is hysteresis comparator. Our response: In order to achieve frequency measurement, we use hysteresis comparison circuit where LM311 plays an important part. LM311 is a low-speed operational amplifier which is exclusively used in voltage comparison, the

typical response time is 200ns. We can use a single supply voltage or a dual power supply. And we have already added the above part to our manuscript.

2. Line61: Please explain how you implement the optocoupler isolation. Our response: HCPL0723 is a high-speed, positive logic CMOS optocoupler of AVAGO TECHNOLO-GIES company, which supports up to 50M bit rate transmission speed, the maximum transmission delay is 22ns, and the external circuit connection is simple, which only needs to add a Coupling capacitors at both ends of the power supply, as shown in Fig. 3. The input and output direction of optocoupler should be confirmed one by one to ensure the correctness of the circuit. Other from traditional optocouplers, the photodiode control pin of HCPL0723 does not flow current, and the output logic pins do not need to be pulled up. Actually, the current that drives the photodiode flows from the input pin (1 pin). And we have already added the above part to our manuscript.

3. Line91: According to your discussion, the atomic clock is a very important part in your device, so in order to help the author better understand the principles of your instrument, it is necessary to elaborate on its circuit structure. Our response: Thanks for your suggestion, we have added the circuit structure of high precision atomic clock to our manuscript. The block diagram of the high-precision atomic clock circuit is shown in Figure 5. Clock board consists of three parts: MSP430 microcontroller, GPS module and atomic clock module. In addition, the clock board also includes some necessary download ports, configration interface and the interface with the main control board.

MSP430 microcontroller , the master, downloads the program through the JTAG download port. The microcontroller communicates with the atomic clock SA.45S through serial port 1, receives the serial data to obtain the atomic clock running status, and sends the serial port command to control its taming time. MSP430 and GPS module communicate with each other through serial port 2. The transmitter of the GPS serial port is also connected to the IO of the FPGA so that the acquisition data contains the location coordinate information. The GPS second pulse signal is connected to the atomic clock of the PPS_IN pin to tame the atomic clock module.
GPS sets the mode of operation through its full-speed USB interface, after setting the work mode of host computer software, you can store the work mode information in the internal of GPS, so that host computer can automatically read the last saved work mode information from the internal of GPS when the module power was cut off.

Please also note the supplement to this comment:
https://www.geosci-instrum-method-data-syst-discuss.net/gi-2017-26/gi-2017-26-SC1-supplement.pdf
* * *
**Fig. 1.**

[Figure]

**Fig. 2.**

---

## Referee Comment (RC2) · Anonymous Referee #3 · 11 Sep 2017

The paper entitled "Development of a full-waveform voltage and current recording device for multichannel transient electromagnetic transmitters" is an interesting original work that explain the construction of device to record a full waveform voltage and current signal from a Multichannel Transient Electromagnetic Transmitter.

The work is within the aim of the GI publication and it presents novel and interesting instrumentation. Abstract summarize properly the contents of the article, and the structure of the paper makes it clear to understand. References are enough in number and quality.

It present interesting results and clearly describes the different parts of the measuring

device. The work is completed presenting not only the hardware but also a software to help in data acquisition and interpretation. They also present test results for the four different channels (high and low speed voltage and high and low speed current channels) and results seems to be satisfactory.

After reading previous discussion I have some questions and suggestions for minor revision:

- Text and labels in figures 6, 7, 8 and 9 are very small. The are difficult to read and understand. Please solve this.

- Software is built in a MS Windows PC with Visual Studio using C# forms. Would it be possible to run in other operating systems (Mac OS, Linux,...)? Some comment about this would be appreciated.

- It is not clear whether if device has been tested out of lab or not. Have the authors done some real-time live measures? If so, some comments about efficiency compared to other methods such as traditional transient electromagnetic method should be done. If not, please ignore this.

---

## Author Comment (AC1) · 13 Sep 2017

To esteemed anonymous referee #3. Your comments impress us a lot. We really appreciate your time and energy for reviewing our manuscript. It is hard for us to express our grateful feeling. In fact, we have learned many things during this revision process, and such experience would be very helpful for our future study. We herewith provide our response to your comments as below:

1. Text and labels in figures 6, 7, 8 and 9 are very small. They are difficult to read and understand. Please solve this. Our response: We have replaced Figure 6, 7, 8, 9 with clearer pictures. The changes we made in manuscript are as follows. 2. Software is

[Figure]

built in a MS Windows PC with Visual Studio using C# forms. Would it be possible to run in other operating systems (Mac OS, Linux,...)? Some comment about this would be appreciated. Our response: Thanks for your suggestion. It is possible to run in other operating systems (Mac OS, Linux,...) as long as users write the appropriate software program.

3. It is not clear whether if device has been tested out of lab or not. Have the authors done some real-time live measures? If so, some comments about efficiency compared to other methods such as traditional transient electromagnetic method should be done. If not, please ignore this. Our response: The device we develop has been tested out of lab, and we have done some real-time live measures. While so far, we can't find a similar device for comparative testing, so there are no comments about efficiency compared to other methods.

Please also note the supplement to this comment:
https://www.geosci-instrum-method-data-syst-discuss.net/gi-2017-26/gi-2017-26-AC1-supplement.pdf
* * *
[Figure]

**Fig. 1.** Figure6

**Fig. 2.** Figure7

[Figure]

**Fig. 3.** Figure8a

[Figure]

**Fig. 4.** Figure8b

[Figure]

**Fig. 5.** Figure9a

[Figure]

**Fig. 6.** Figure9b

**Supplement:**

**Point-by-point replies to the comments of Referee #3**

**To esteemed anonymous referee #3.**

Your comments impress us a lot. We really appreciate your time and energy for reviewing our manuscript. It is hard for us to express our grateful feeling. In fact, we have learned many things during this revision process, and such experience would be very helpful for our future study. We herewith provide our response to your comments as below:

1. Text and labels in figures 6, 7, 8 and 9 are very small. They are difficult to read and understand. Please solve this.

    ***Our response:***

    We have replaced Figure 6, 7, 8, 9 with clearer pictures. The changes we made in manuscript are as follows.

[Figure]

Figure 6: Block diagram of the Bluetooth and 485 module transmission.

[Figure]

Figure 7: Interface display of the Main Window form on the upper computer.

[Figure]

a. Time domain waveform of the 20-Vpp, 100-Hz sinusoid acquired by the recording device at a 1000-V level

[Figure]

b. Amplitude-frequency characteristics of the 32000 samples subjected to FFT (Hanning window truncation)

Figure 8: Test results of the low-speed voltage acquisition channel.

[Figure]

a. Time domain waveform of the 20-Vpp, 100-kHz sinusoid acquired by the high-speed voltage recording device at a 100-V level

[Figure]

b. Amplitude-frequency characteristics of the 4906 samples subjected to FFT (Hanning window truncation)

Figure 9: Test results of the high-speed voltage acquisition channel.

2. Software is built in a MS Windows PC with Visual Studio using C# forms. Would it be possible to run in other operating systems (Mac OS, Linux,...)? Some comment about this would be appreciated.

   *Our response:*

   Thanks for your suggestion. It is possible to run in other operating systems (Mac OS, Linux,...) as long as users write the appropriate software program.

3. It is not clear whether if device has been tested out of lab or not. Have the authors done some real-time live measures? If so, some comments about efficiency compared to other methods such as traditional transient electromagnetic method should be done. If not, please ignore this.

   *Our response:*

The device we develop has been tested out of lab, and we have done some real-time live measures. While so far, we can't find a similar device for comparative testing, so there are no comments about efficiency compared to other methods.